# AI-based Japanese Short-answer Scoring and Support System

## Abstract

We have developed an automated Japanese short-answer scoring and support machine for new National Center written test exams. Our approach is based on the fact that recognizing textual entailment and/or synonymy has been almost impossible for several years. The system generates automated scores on the basis of evaluation criteria or rubrics, and human raters revise them. The system determines semantic similarity between the model answers and the actual written answers as well as a certain degree of semantic identity and implication. Owing to the need for the scoring results to be classified at multiple levels, we use random forests to utilize many predictors effectively rather than use support vector machines. An experimental prototype operates as a web system on a Linux computer. We compared human scores with the automated scores for a case in which 3–6 allotment points were placed in 8 categories of a social studies test as a trial examination. The differences between the scores were within one point for 70–90 percent of the data when high semantic judgment was not needed.

## 1 Introduction

Educational advisory body to the Japanese government has decided that writing tests will be introduced into the new national center test for university entrance examinations, as announced in a final report (MEXT, 2016) at the high school and university articulation meeting by the Ministry of Education, Culture, Sports, Science and Technology. The use of AI-based computers was proposed to stabilize the test scores efficiently. The required type of writing test is a short-answer test, where a correct answer is expected to exist. Therefore, the test is scored by judging agreement of the meaning with the correct answer. Another type of writing test is essay writing, where a correct answer does not exist. The written answers are evaluated based on the rhetoric, the connection expressions, and the content.

Because short-answer scoring involves technical difficulty, the number of characters is restricted to 80 characters at most from dozens of characters. Two characters in Japanese are generally equivalent to one word in English. A short-answer test is widely considered to be more authentic and reliable for measuring ability compared with a multiple-choice test. If technical problems related to the short-answer test are solved, the potential demand for its use, as well as that for the national center test, will be enormous. Many systems for evaluating essays have been developed and offered in the United States (Shermis and Burstein, 2013). The authors' group also developed the first and most well-known Japanese automated essay scoring system named Jess (Ishioka and Kameda, 2006), and it is in practical use now.

While a short-answer scoring system has been developed because of its importance, various technical problems remain unsolved. New York University (NYU) and the Educational Testing Service (ETS) developed the first automated scoring tools in this field; they evaluated the NYU online program (Vigilante, 1999). Leacock and Chodorow (2003) reported the latest specifications of the c-rater developed by ETS. Pulman and Sukkarieh (2005) tried to generate several sentences having the same meaning as the correct answer sentence using the natural language technique of information extraction. However, the concordance rate with human examiners was found to be small and impractical.

The key technology to solve the short answer scoring seems to be recognizing textual entailment. The National Institute of Informatics (NII) in Japan has promoted the "Todai robot project" to solve multiple choice tests of the Japanese national center test for university entrance examinations (NCTUEE) using this technology with knowledge resources from many textbooks and Wikipedia. They found that perfect recognition of textural entailment or correct understanding of the meaning is difficult at present (NII, 2013). To combine several methods and patch them ad hoc is the most that can be done if people feigned to answer. We have also tried to acquire the basic technology and to produce an automated written short-answer system based on our scientific research conducted in 2014–2016, but we reached the limit of the system.

Therefore, we thought of a support system for short written tests where a human rater can correct the automated score by referring to the original scores. When the human rater agrees with the result of the automated score, he/she can just approve the score indicated by default and can produce the corresponding mark. We chose to leave room for human raters to overwrite it without making it a perfect automated scoring system.

The new examination, which was created by the NCTUEE, will utilize the written test for Japanese literature, whose scoring seems to be more difficult than the examinations of science and social studies, which are prepared basically using the facts written in their respective textbooks. The written test for Japanese literature needs reading skills rather than skills of information processing and pattern matching. In this test in particular, we have to detect the semantic difference in the compared written answers with the model sentences, though almost no difference exists in the vocabulary used. Therefore, we tried to tackle the scoring of short sentences in social studies, where precise judgments are less needed.

In section 2, we indicate the test items and the model answers used in a trial examination for university entrance examinations. In section 3, we show the specifications of our proposed system. In section 4, we present our evaluation of the performance on eight tests of social studies. Section 5 concludes with a summary.

## 2 Test items used in a trial examination

We assigned a theme in three subjects of world history, Japanese history, and geography of the "Gakken nation-wide trial examination" in fiscal year 2015. The world history test set includes four written test items and two test items each for geography and Japanese history; the total is 8 test items.

Table 1 shows the "content" asked and the "correct answer," which are given to test examinees in a distributed booklet of "test answers and explanations."

Herein, I comment a little about the correct answer of World history B2 #3. "Jizya" was the capitation imposed on non-Muslims as compensation required for their beliefs in the Islamic world; when indigenous people in a place of conquest converted to Islam, they were not taxed originally. However, "kharaj" was a land tax, and it was imposed on those who possessed land, regardless of whether or not they were Muslim.

However, even if indigenous people converted to Islam in the time of Umayyad, "Jizya" was still imposed, and even if land was possessed, they were exempted from "kharaj." The Abbasid dynasty returned it to the original state (Table 2). This test item asks about the content, and the recognizing correct meaning is necessary for getting scores.

Table 2: Tax system changes from Umayyad to Abbasid dynasty

|  | Muslims | non-Muslims |
|---|---|---|
| Jizya | tax → tax exempt | tax |
| kharaj | tax | tax exempt → tax |

## 3 Specifications of the scoring support system

### 3.1 Outline

Our system is for automated scoring and for supporting human raters. The approach functions as follows.

1. A system automatically judges each answer posed on whether or not its prepared key phrases agree with those of the model answer using the "scoring criteria" from a surface-like point of view.

Table 1: The content and the correct answer examples of written test items

| Subject | Test Item # (Allotment) | Content and the correct answer |
|---|---|---|
| World history B2 | #1 (3 pt.) | **[content]** Ancient Greece: Solon's politics of assets **[correct answer]** A citizen's right to vote was set according to oneEfs political classes, which depended on his or her ownership of properties. (17 characters in Japanese) |
| | #3 (5 pt.) | **[content]** Islam: Tax system in the Abbasid dynasty **[correct answer]** When indigenous people in a place of conquest were Muslim, they were exempted from "Jizya." When Arabs had land in a place of conquest, "kharaj" was imposed. (60 chars.) |
| Japanese history B1 | #2 (3 pt.) | **[content]** Peace treaty of the Russo-Japanese War; Change in the territory **[correct answer]** The south in Sakhalin was ceded to Japan from Russia. (22 chars) |
| | #4 (5 pt.) | **[content]** Dissolution of financial giants after World War II **[correct answer]** A holding company, a cartel, and a trust were prohibited by the Antimonopoly Act, and a huge monopoly was divided by the excessive economic power decentralization law. (59 chars) |
| Japanese history B2 | #1 (3pt.) | **[content]** The Emperor Genmei: Tax burden in the ordinance system **[correct answer]** A certain amount of cloth is offered instead of labor for capital[m9]. (20 chars) |
| | #3 (6 pt.) | **[content]** Kamakura era second half: Commercial activities in the city **[correct answer]** The carrier, called a "Toi," and the usury person, called a "Kariage," appeared, and an exchange settled in the bill was started instead of sending money. (59 chars) |
| Geometry B | #1 (3 pt.) | **[content]** The genesis of the Namib desert **[correct answer]** The cold current that flows through an offing[m10] and a medium latitude high-pressure area (19 chars) |
| | #4 (6 pt.) | **[content]** The population of the world: A comparison in India and China **[correct answer]** The birthrate in India is higher than that of China, and the population growth is remarkable because India failed in its attempts at strict birth control, whereas China conducted a one-child policy. (59 chars) |

©Gakken Holdings

2. The system gives not only a temporary score based on the criterion-based judgment but also a prediction score offered by machine learning based on the understanding of other human raters or supervised data. A certain degree of semantic meaning is also used.

3. A human rater can certify the prediction score by which a system presents this information as reference. He or she can correct this and overwrite based on his/her judgment.

To reduce the time and effort, the system precision should possess a certain degree of fitness with human ratings; more than 80% of the precision is desirable for tentative targets.

The flowchart of our system is as shown in figure 1.

(a) Before scoring, we collected a lot of score data from various human raters and performed a machine learning of "Random Forests" (Breiman, 2001). The degree of fitness with the scoring guideline is also necessary. On the basis of these learning results, we set up a scoring engine to return the scores for new answers.

(b) The system generates a scoring screen written in the Hyper Text Markup Language.

(c) A user or human rater opens a scoring screen of (b) using a web browser on his/her terminal machine. Then, a CGI program is activated. The recommended value as a result of the scoring engine of (a) is indicated here. The scoring result is stocked in a file or a database. The user repeats this mark operation.

### 3.2 Scoring Screen

Figure 2 shows a screen shot of our prototype system. "The answer sentence that should be scored" (in red ink) is located in the upper part of the system; the middle part has some scoring criteria such as "synonyms and permitted different transcriptions," "model or correct answers that warrant a full mark," "partial phrases that warrant partial scores," and "mandatory phrases." For the "model answer" and "partial correct phrases," the system judges the degree of fitness with the answer sentence to be scored; the system also judges whether or not the answer sentence includes "mandatory phrases," whether or not it is meaningfully composed, and whether or not it exceeds the character limit; if the answer should be written as a noun or noun phrase, the system judges whether or not it matches the specified "type" format. These judg-

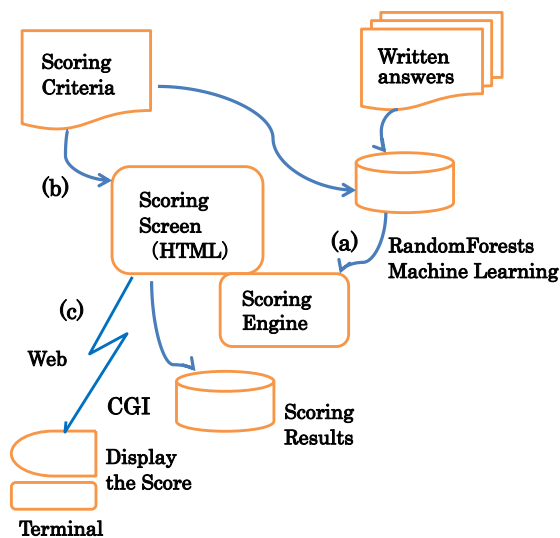

Figure 1: Flowchart of the system

ments are either yes or no, and toggle buttons are used. A human rater reviews these judgments and revises them if necessary.

Tentative scores located in the lower part are based on the aforementioned alternative judgment. The right-hand window is to determine the final score. The initial mark is settled by which predictive probability based on the past learned results gives the maximum. The probability values are also indicated.

When no learning data exist, that is to say, when no pre-scored data about the relevant test item exist, the message to that effect is shown in the top windows: no probability and no initial mark are naturally determined.

### 3.3 Difference between the tentative score and the mechanical prediction score

The example indicated by figure 2 is a case of World history B2 #3. The answer indicated here is as follows: Indigenous people in a place of conquest originally had both "Jizya" and "kharaj" imposed on them, but only "kharaj," like it was with Arabs, later came to be imposed on them. Compared with the model answer of "Even when indigenous people in a place of conquest were Muslim, they were exempted from Jizya. When an Arab had land in a place of conquest, kharaj was imposed," we found that the appearance of written words was similar, but the apparent meaning of the sentences was quite different. Therefore,

the system gives a score of 4 points of the tentative score (5 points of allotment) determined by buttons checked based on agreement of surface-like words and phrases that appeared; but the mechanical prediction score is 0 points, and it takes account of other elements besides the surface-like side. The prediction probability of the recommended score is 0.79. It shows that the effect of the machine learning is functioning appropriately.

### 3.4 Automatic screen creation from a scoring criterion file

Our system is a Web application. Thus, the screen indicated by figure 2 is generated by HyperText Markup Language. We built the mechanism to make this HTML file automatically from a plain scoring criterion file that a computer beginner can handle.

Figure 3 is a plain original file that makes a screen like the one in figure 2. Two or three elements are set for criteria. In order, the label, allotment of points, and correspondence are located. The tab is the delimiter.

Synonyms and different transcriptions are recorded in "syno," which appeared in "gold" as a model answer and in "part" as a partially correct phrase. "Syno" is not always limited to a definite lexical meaning. When it has semantically the same meaning, it is also permitted. "Part" includes two types; one is possible to add to a partial point, and the other is for which a maximum is taken. If there are multiple same labels (for example, part1), we use the maximum of the points; different labels (for example, part 1 and part 2) can add the allotted points. "Lack" is a mandatory phrase; if no phrases exist, the point is deducted. A comma can be used for the meaning of "both." "Vol" shows the number of characters available. "None" shows a nonsense sentence, and "goji" shows a wrong word such as kanji that does not exist. Minus points indicate points to be deducted.

We use "fitness" as the degree of the relationship between the written answer and "model answer" designated in "gold" or "partial correct phrases" in "part." We define this as the harmonic mean of two kinds of relationships: one is the degree of the reference during the sentence keywords from the viewpoint of a written answer; the other is that from a model answer. These relationships are just like precision and recall often used in in-

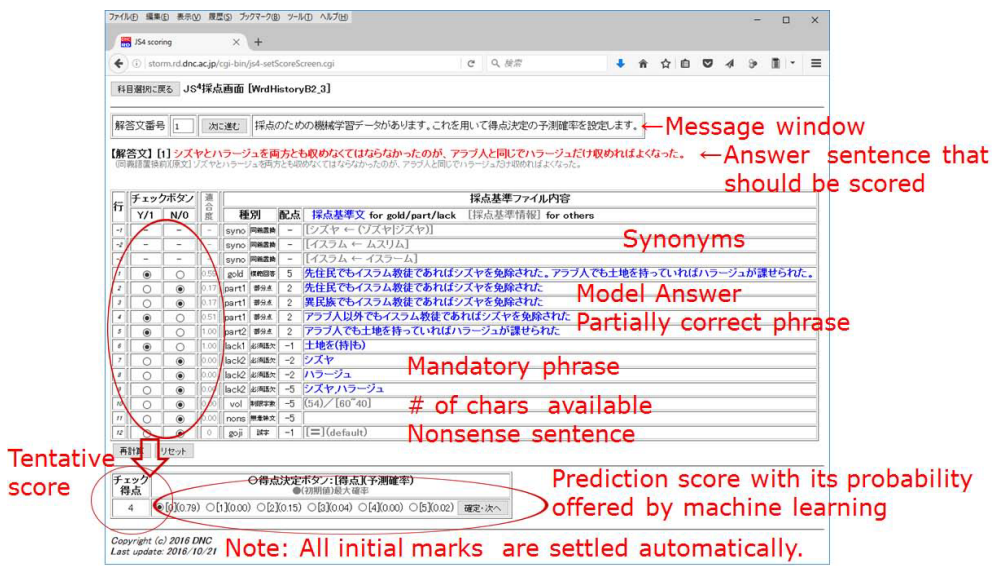

Figure 2: Short-answer scoring and support system screen (In case of world history B2 #3)

```
syno    jizya    jizyah
syno    Muslim   Islam
syno    "Indigenous people" "Different ethnic groups"
gold    5    "Even when indigenous people in a place of conquest were Muslim, they
 were exempted from Jizya. Even when an Arab had land in a place of conquest, kharaj
 was imposed."
part1   2    "Even when indigenous people in a place of conquest were Muslim, they
 were exempted from Jizya."
part2   2    "Even when an Arab had land in a place of conquest, kharaj was imposed."
lack1  -1    "had land"
lack2  -2    jizya
lack2  -2    kharaj
lack2  -5    jizya,kharaj
vol    -5    60-40
nons   -5
goji   -1
```

Figure 3: Scoring criterion file (labels, allotment of points, and correspondences are tab delimited.)

formation retrieval, e.g., a Google search. This harmonic mean or "fitness" is called an F-measure taking a float number from 0 to 1. Our system rounds this to either 0 or 1 as a toggle button occurrence, and it shows a non-rounded value as a reference for the user.

## 4 Performance Evaluation

### 4.1 Evaluation of the classification

We built a prediction model using machine learning of random forests (RFs) V4.1 (Liaw and Wiener, 2015; Breiman and Cutler, 2004). The predictors include not only the degree of fitness indicated in Figure 3 but also semantic (cosine) similarity between the answer, model answer, and test item sentences. The reasons for using the RFs in the methods of many machine learning techniques are as follows:

1. When using many predictor variables, the classification often functions effectively.

2. The degree of contributions can be estimated to determine effective predictor variables quantitatively in the classification.

3. RFs are suitable for this because test scoring requires multiple classifications with values of 0–3 or 0–6.

For eight test items indicated in table 1, we compared human ratings with the estimate based prediction model. The best rate that that a prediction and a correct answer were identical is 78% in the cases of Geometry B #4. The worst rate is 43% in the cases of Japanese History B1 #2, which was surprisingly low. This is because about 80% of examinees got zero scores; thus, machine learning does not work well. We omitted cross matrices between human ratings and the estimate because of space limitations.

Table 3 shows the probability in which the differences between the scores were within one point. These values consist of 71–95% removing the case of world history B2 #3, which is necessary for correct understanding of the meaning. It shows the performance of the classification was in the level available.

RFs do not need cross validation to calculate the error rate. That is, a separated test set is not necessary to determine it (Breiman, 2001). The error rate can be estimated internally during a run. In RFs, each tree is constructed using a different

Table 3: Probability in which the estimates differed from the human ratings within one point

| Item # | Prob. | Item # | Prob. |
|---|---|---|---|
| World History B2 #1 | 0.75 | Jpn History B2 #1 | 0.86 |
| World History B2 #3 | 0.48 | Jpn History B2 #3 | 0.71 |
| Jpn History B1 #2 | 0.76 | Geo B #1 | 0.91 |
| Jpn History B1 #4 | 0.88 | Geo B #4 | 0.95 |

bootstrap sample from two-thirds of the original data. The remaining one-third of cases is used for the test data.

The default procedure runs 500 times and forms the final classification tree using the most votes, when using the RandomForest package implemented in R(Liaw and Wiener, 2015). The error rate for the test set can be obtained at the same time. The numerical values shown in Table 3 were obtained using this procedure. The sample sizes ranged in 70–120, depending on the subjects.

If we compare the original data with the estimates using the final RF model, the concordance rate will be near 100% because too many predictive variables were compared with the sample size.

### 4.2 Variables that contribute to the classification

RFs evaluate the importance of variables in distinction using an index of the Gini coefficient. The bigger the coefficient, the more the classification is affected. Figure 4 shows variables arranged in descending order of the Gini coefficient; the horizontal axis shows their values. Due to space limitations, we show three cases of world history B2 #3, Japanese history B1 #2, and geometry B #4.

The three dominant contributions in case of world history B2 #3 are as follows:

**QA_sim:** The cosine similarity between test item sentences and the answer.

**SA_sim_1:** The cosine similarity between model answer #1 and the answer.

**sa_jpkwrel_Fvl_gold__std01:** The F-measure in keywords agreement between model answer #1 and the answer.

A variable name that starts with a capital letter implies linguistic semantic meaning built by the vocabulary used in Japanese Wikipedia. The typical examples are cosine similarity, precision, recall, and F-measure in the semantic space. Variations that multiply the allotment are also included.

The variable names that start with a capital letter are an index showing surface-like lexical relationships. The total number of used variables is from 40 to 70, depending on the size of the scoring criteria.

When the indexes indicated in figure 4 are compared by each test item, the variables that contribute to distinction are not fixed. However, the F-measure and cosine similarity between the model (correct) answer and the answer in the semantic space often appear in a higher position. Generally speaking, semantic variables are dominant compared with surficial lexical variables.

### 4.3 Inspection using agreement of case elements

The case grammar proposed by Charles J. Fillmore (Fillmore, 1968) is part of a theory that tries to understand sentence meanings by putting a verb (the word with declined or conjugated endings) at the center of the understanding and that analyzes using a combination with deep cases such as the Agent and Object. Deep cases are often not determined easily, so we use surface cases instead of them. We tried a check on the agreement with case elements of a "model answer" and those of a "written answer." The mechanism for the inspection had been arranged.

However, almost no agreement of case elements happened; the degree of agreement does not become one of the effective predictors of the classification. The ways in which sentences can be filled out to indicate the same meaning are numerous. Therefore, we did not add case grammar variables on the agreement of the case elements in this experiment.

## 5 Conclusion

Recognizing textual entailment between a model (correct) answer and a written answer is still difficult technically because complicated collation needs to be determined under contractual and semantic levels. Our technique is based on the collation between the keywords in two answers, and it uses both predictors considering superficial and semantic aspects. Therefore, it can be judged as sufficiently realistic for an approach of the first step. A form that entrusts the last judgment to the human is most suitable.

In this study, we investigated cases of scoring in social studies (geography and world/Japanese

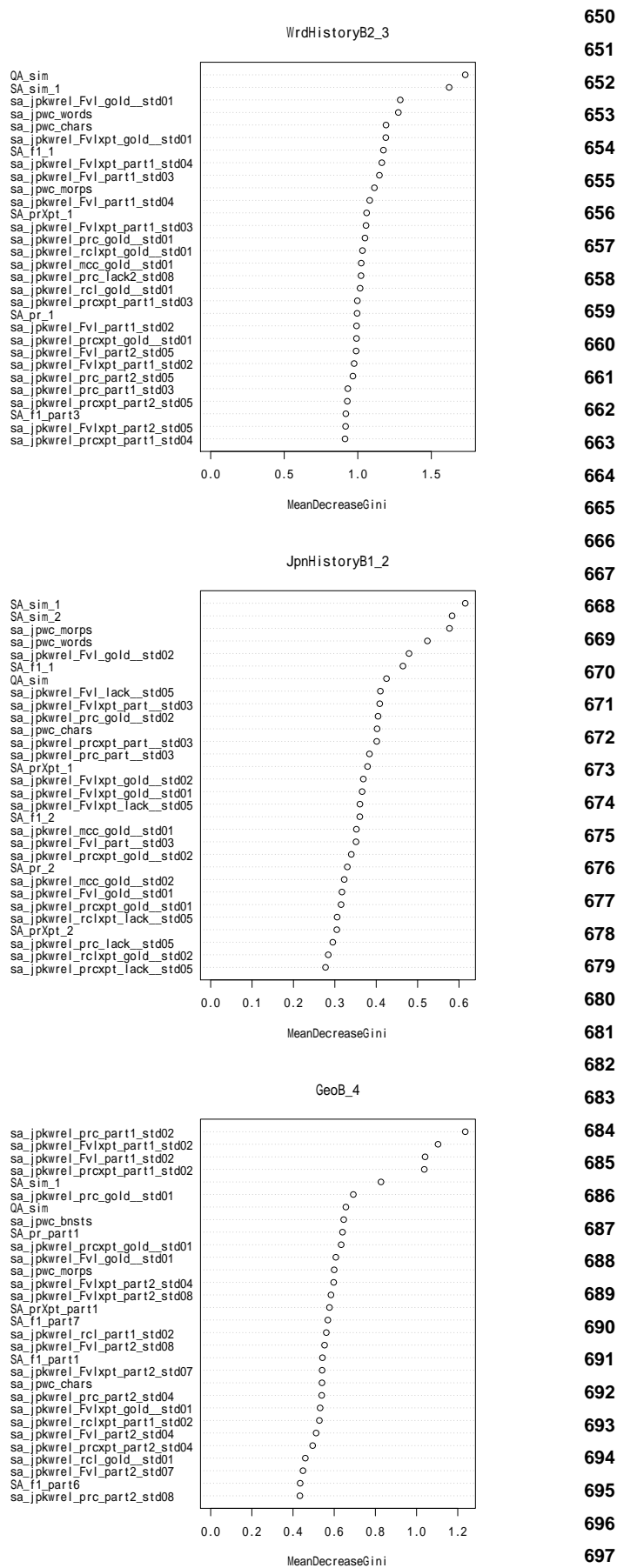

Figure 4: Variables that contribute to distinction (In the case of world history B2 #3, Japanese history B1 #2, and geometry B #4)

history). Our procedure will be applied to other subjects, such as Japanese literature, when many different transcriptions of a correct answer are prepared. Many different expressions in the same sense can be allowed, especially in Japanese. Our system has a mechanism to choose the biggest score among the same labels, so this can be prepared using the specifications of the current state.

Moreover, when there is a content-like important word or phrase such as "father's feeling," our system can register this type of words as "mandatory" in scoring criteria and can decrease the score by a suitable allotment when it is lacking. This function has already been implemented and will be helpful to understand the semantics.

However, a sufficiently large number of human scores cannot be provided for supervised learning. Indeed, actual written answer scores are often zero because they are illogical or are off-topic. Obtaining suitable and well-balanced score data will be necessary to ensure proper estimation.

Because a short written test has been proposed as a new common test for entering Japanese universities, our new scoring and support system is now being considered. We hope this system will provide researchers who study this field and practical businesspeople with useful information and suggestions.

## Acknowledgments

The authors would like to thank everybody in the Gakken group for offering the test sets, scoring criteria, and human scores. In particular, Mr. Tsugunao Matsuoka and Ms. Naoko Saigo gave us valuable comments. This project was supported by JSPS KAKENHI Grant Number JP:26350357.

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
