# Peer review of "AI-based Japanese Short-answer Scoring and Support System"

_ACL 2017 — decision unknown_

[Official Review · Reviewer 1 · rating 1 · confidence 4]
soundness 4 · originality 3 · clarity 2 · impact 2 · substance 1 · appropriateness 4 · meaningful comparison 2 · presentation format Poster

This paper describes a system to assist written test scoring.

- Strengths:
The paper represents an application of an interesting NLP problem --
recognizing textual entailment -- to an important task -- written test scoring.

- Weaknesses:
There isn't anything novel in the paper. It consist of an application of an
existing technology to a known problem.

The approach described in the paper is not autonomous -- it still needs a human
to do the actual scoring. The paper lacks any quantitative or qualitative
evaluation of how useful such system is. That is, is it making the job of the
scorer easier? Is the scorer more effective as compared to not having automatic
score?

The system contains multiple components and it is unclear how the quality of
each one of them contributes to the overall experience.

The paper needs more work with the writing. Language and style is rough in
several places.

The paper also contains several detailed examples, which don't necessarily add
a lot of value to the discussion.

 For the evaluation of classification, what is the baseline of predicting the
most frequent class?

- General Discussion:
I find this paper not very inspiring. I don't see the message in the paper
apart from announcing having build such a system

[Official Review · Reviewer 2 · rating 2 · confidence 4]
soundness 4 · originality 3 · clarity 2 · impact 3 · substance 2 · appropriateness 3 · meaningful comparison 1 · presentation format Poster

- Strengths:

This paper tries to tackle a very practical problem: automated short answer
scoring (SAS), in particular for Japanese which hasn't gotten as much attention
as, say, English-language SAS.

- Weaknesses:

The paper simply reads like a system description, and is light on experiments
or insights. The authors show a lack of familiarity with more recent related
work (aimed at English SAS), both in terms of methodology and evaluation. Here
are a couple:

https://www.aclweb.org/anthology/W/W15/W15-06.pdf#page=97
https://www.aclweb.org/anthology/N/N15/N15-1111.pdf

There was also a recent Kaggle competition that generated several
methodologies:

https://www.kaggle.com/c/asap-sas

- General Discussion:

To meet ACL standards, I would have preferred to see more experiments (feature
ablation studies, algorithm comparisons) that motivated the final system
design, as well as some sort of qualitative evaluation with a user study of how
the mixed-initiative user interface features led to improved scores. As it is,
it feels like a work in progress without any actionable new methods or
insights.

Also, Pearson/Spearman correlation and kappa scores are considered more
appropriate than accuracy for these sorts of ordinal human scores.

[Official Review · Reviewer 3 · rating 3 · confidence 3]
soundness 4 · originality 3 · clarity 3 · impact 3 · substance 3 · appropriateness 4 · meaningful comparison 2 · presentation format Poster

This paper presents a text classification method based on pre-training
technique using both labeled and unlabeled data. The authors reported
experimental results with several benchmark data sets including TREC data, and
showed that the method improved overall performance compared to other
comparative methods.

I think the approach using pre-training and fine-tuning itself is not a novel
one, but the originality is the use of both labeled and unlabeled data in the
pre-training step. 
The authors compare their results against three baselines, i.e. without
pre-training and a deep learning with unsupervised pre-training using deep
autoencoders, but I think that I would be interesting to compare the method
against other methods presented in the introduction section.